# State-wise Constrained Policy Optimization

## Abstract

Reinforcement Learning (RL) algorithms have shown tremendous success in simulation environments, but their application to real-world problems faces significant challenges, with safety being a major concern. In particular, enforcing state-wise constraints is essential for many challenging tasks such as autonomous driving and robot manipulation. However, existing safe RL algorithms under the framework of Constrained Markov Decision Process (CMDP) do not consider state-wise constraints. To address this gap, we propose State-wise Constrained Policy Optimization (SCPO), the first general-purpose policy search algorithm for state-wise constrained reinforcement learning. SCPO provides guarantees for state-wise constraint satisfaction in expectation. In particular, we introduce the framework of Maximum Markov Decision Process, and prove that the worst-case safety violation is bounded under SCPO. We demonstrate the effectiveness of our approach on training neural network policies for extensive robot locomotion tasks, where the agent must satisfy a variety of state-wise safety constraints. Our results show that SCPO significantly outperforms existing methods and can handle state-wise constraints in high-dimensional robotics tasks.

## 1 Introduction

Reinforcement learning (RL) has achieved remarkable progress in games and control tasks [Mnih et al., 2015, Vinyals et al., 2019, Brown and Sandholm, 2018, He et al., 2022, Zhao et al., 2019]. However, one major barrier that limits the application of RL algorithms to real-world problems is the lack of safety assurance. RL agents learn to make reward-maximizing decisions, which may violate safety constraints. For example, an RL agent controlling a self-driving car may receive high rewards by driving at high speeds but will be exposed to high chances of collision. Although the reward signals can be designed to penalize risky behaviors, there is no guarantee for safety. In other words, RL agents may sometimes prioritize maximizing the reward over ensuring safety, which can lead to unsafe or even catastrophic outcomes [Gu et al., 2022].

Emerging in the literature, safe RL aims to provide safety guarantees during or after training. Early attempts have been made under the framework of constrained Markov Decision Process, where the majority of works enforce cumulative constraints or chance constraints [Ray et al., 2019, Achiam et al., 2017a, Liu et al., 2021]. In real-world applications, however, many critical constraints are instantaneous. For instance, collision avoidance must be enforced at all times for autonomous cars [Zhao et al., 2023]. Another example is that when a robot holds a glass, the robot can only release the glass when the glass is on a stable surface. The violation of those constraints will lead to irreversible failures of the task. In this work, we focus on state-wise (instantaneous) constraints.

The State-wise Constrained Markov Decision Process (SCMDP) is a novel formulation in reinforcement learning that requires policies to satisfy hard state-wise constraints. Unlike cumulative or probabilistic constraints, state-wise constraints demand full compliance at each time step as formalized by [Zhao et al. [2023]]. Existing state-wise safe RL methods can be categorized based on

whether safety is ensured during training. There is a fundamental limitation that it is impossible to guarantee hard state-wise safety during training without prior knowledge of the dynamic model. The best we can achieve in a model free setting is to learn to satisfy the constraints using as few samples as possible, which is the focus of this paper. We aim to provide theoretical guarantees on state-wise safety violation and worst case reward degredation during training.

Our approach is underpinned by a key insight that constraining the maximum violation is equivalent to enforcing state-wise safety. This insight leads to a novel formulation of MDP called the *Maximum Markov Decision Process* (MMDP). With MMDP, we establish a new theoretical result that provides a bound on the difference between the maximum cost of two policies for episodic tasks. This result expands upon the cumulative discounted reward and cost bounds for policy search using trust regions, as previously documented in literature [Achiam et al., 2017b]. We leverage this result to design a policy improvement step that not only guarantees worst-case performance degradation but also ensures state-wise cost constraints. Our proposed algorithm, *State-wise Constrained Policy Optimization* (SCPO), approximates the theoretically-justified update, which achieves a state-of-the-art trade-off between safety and performance. Through experiments, we demonstrate that SCPO effectively trains neural network policies with thousands of parameters on high-dimensional simulated robot locomotion tasks; and is able to optimize rewards while enforcing state-wise safety constraints. This work represents a significant step towards developing practical safe RL algorithms that can be applied to many real-world problems.

# 2 Related Work

## 2.1 Cumulative Safety

Cumulative safety requires that the expected discounted return with respect to some cost function is upper-bounded over the entire trajectory. One representative approach is constrained policy optimization (CPO) [Achiam et al., 2017a], which builds on a theoretical bound on the difference between the costs of different policies and derives a policy improvement procedure to ensure constraints satisfaction. Another approach is interior-point policy optimization (IPO) [Liu et al., 2019], which augments the reward-maximizing objective with logarithmic barrier functions as penalty functions to accommodate the constraints. Other methods include Lagrangian methods [Ray et al., 2019] which use adaptive penalty coefficients to enforce constraints and projection-based constrained policy optimization (PCPO) [Yang et al., 2020a] which projects trust-region policy updates onto the constraint set. Although our focus is on a different setting of constraints, existing methods are still valuable references for illustrating the advantages of our SCPO. By utilizing MMDP, SCPO breaks the conventional safety-reward trade-off, which results in stronger convergence of state-wise safety constraints and guaranteed performance degradation bounds.

## 2.2 State-wise Safety

**Hierarchical Policy**   One way to enforce state-wise safety constraints is to use hierarchical policies, with an RL policy generating reward-maximizing actions, and a safety monitor modifying the actions to satisfy state-wise safety constraints. Such an approach often requires a perfect safety critic to function well. For example, conservative safety critics (CSC) [Bharadhwaj et al., 2020] propose a safe critic $Q_C(s,a)$, providing a conservative estimate of the likelihood of being unsafe given a state-action pair. If the safety violation exceeds a predefined threshold, a new action is re-sampled from the policy until it passes the safety critic. However, this approach is time-consuming. On the other hand, optimization-based methods such as gradient descent or quadratic programming can be used to find a safe action that satisfies the constraint while staying close to the reference action. Unrolling safety layer (USL) [Zhang et al., 2022a] follows a similar hierarchical structure as CSC but performs gradient descent on the reference action iteratively until the constraint is satisfied based on learned safety critic $Q_C(s,a)$. Finally, instead of using gradient descent, Lyapunov-based policy gradient (LPG) [Chow et al., 2019] and SafeLayer [Dalal et al., 2018] directly solve quadratic programming (QP) to project actions to the safe action set induced by the linearized versions of some learned critic $Q_C(s,a)$. All these approaches suffer from safety violations due to imperfect critic $Q_C(s,a)$, while those solving QPs further suffer from errors due to the linear approximation of the critic. To avoid those issues, we propose SCPO as an end-to-end policy which does not explicitly maintain a safety monitor.

**End-to-End Policy** End-to-end policies maximize task rewards while ensuring safety at the same time. Related work regarding state-wise safety after convergence has been explored recently. Some approaches [Liang et al., 2018, Tessler et al., 2018] solve a primal-dual optimization problem to satisfy the safety constraint in expectation. However, the associated optimization is hard in practice because the optimization problem changes at every learning step. Bohez et al. [2019] approaches the same setting by augmenting the reward with the sum of the constraint penalty weighted by the Lagrangian multiplier. Although claimed state-wise safety performance, the aforementioned methods do not provide theoretical guarantee and fail to achieve near-zero safety violation in practice. He et al. [2023] proposes AutoCost to automatically find an appropriate cost function using evolutionary search over the space of cost functions as parameterized by a simple neural network. It is empirically shown that the evolved cost functions achieve near-zero safety violation, however, no theoretical guarantee is provided, and extensive computation is required. FAC [Ma et al., 2021] does provide theoretically guaranteed state-wise safety via parameterized Lagrange functions. However, FAC replies on strong assumptions and performs poorly in practice. To resolve the above issues, we propose SCPO as an easy-to-implement and theoretically sound approach with no prior assumptions on the underlying safety functions.

# 3 Problem Formulation

## 3.1 Preliminaries

In this paper, we are especially interested in guaranteeing safety for episodic tasks, which falls within in the scope of finite-horizon Markov Decision Process (MDP). An MDP is specified by a tuple $(\mathcal{S}, \mathcal{A}, \gamma, R, P, \mu)$, where $\mathcal{S}$ is the state space, and $\mathcal{A}$ is the control space, $R : \mathcal{S} \times \mathcal{A} \mapsto \mathbb{R}$ is the reward function, $0 \leq \gamma < 1$ is the discount factor, $\mu : \mathcal{S} \mapsto \mathbb{R}$ is the initial state distribution, and $P : \mathcal{S} \times \mathcal{A} \times \mathcal{S} \mapsto \mathbb{R}$ is the transition probability function. $P(s'|s, a)$ is the probability of transitioning to state $s'$ given that the previous state was $s$ and the agent took action $a$ at state $s$. A stationary policy $\pi : \mathcal{S} \mapsto \mathcal{P}(\mathcal{A})$ is a map from states to a probability distribution over actions, with $\pi(a|s)$ denoting the probability of selecting action $a$ in state $s$. We denote the set of all stationary policies by $\Pi$. Subsequently, we denote $\pi_\theta$ as the policy that is parameterized by the parameter $\theta$.

The standard goal for MDP is to learn a policy $\pi$ that maximizes a performance measure $\mathcal{J}_0(\pi)$ which is computed via the discounted sum of reward:

$$\mathcal{J}_0(\pi) = \mathbb{E}_{\tau \sim \pi} \left[ \sum_{t=0}^{H} \gamma^t R(s_t, a_t, s_{t+1}) \right], \tag{1}$$

where $H \in \mathbb{N}$ is the horizon, $\tau = [s_0, a_0, s_1, \cdots]$, and $\tau \sim \pi$ is shorthand for that the distribution over trajectories depends on $\pi : s_0 \sim \mu, a_t \sim \pi(\cdot|s_t), s_{t+1} \sim P(\cdot|s_t, a_t)$.

## 3.2 State-wise Constrained Markov Decision Process

A constrained Markov Decision Process (CMDP) is an MDP augmented with constraints that restrict the set of allowable policies. Specifically, CMDP introduces a set of cost functions, $C_1, C_2, \cdots, C_m$, where $C_i : \mathcal{S} \times \mathcal{A} \times \mathcal{S} \mapsto \mathbb{R}$ maps the state action transition tuple into a cost value. Analogous to (1), we denote

$$\mathcal{J}_{C_i}(\pi) = \mathbb{E}_{\tau \sim \pi} \left[ \sum_{t=0}^{H} \gamma^t C_i(s_t, a_t, s_{t+1}) \right] \tag{2}$$

as the cost measure for policy $\pi$ with respect to cost function $C_i$. Hence, the set of feasible stationary policies for CMDP is then defined as follows, where $d_i \in \mathbb{R}$:

$$\Pi_C = \{\pi \in \Pi \big| \forall i, \mathcal{J}_{C_i}(\pi) \leq d_i\}. \tag{3}$$

In CMDP, the objective is to select a feasible stationary policy $\pi_\theta$ that maximizes the performance measure:

$$\max_{\pi} \mathcal{J}_0(\pi), \ \text{s.t.} \ \pi \in \Pi_C. \tag{4}$$

In this paper, we are interested in a special type of CMDP where the safety specification is to persistently satisfy a hard cost constraint **at every step** (as opposed to cumulative costs over trajectories), which we refer to as *State-wise Constrained Markov Decision Process* (SCMDP). Like CMDP, SCMDP uses the set of cost functions $C_1, C_2, \cdots, C_m$ to evaluate the instantaneous cost of state action transition tuples. Unlike CMDP, SCMDP requires the cost for every state action transition to satisfy a hard constraint. Hence, the set of feasible stationary policies for SCMDP is defined as

$$\bar{\Pi}_C = \{\pi \in \Pi | \forall i, \ \mathbb{E}_{(s_t, a_t, s_{t+1}) \sim \tau, \tau \sim \pi} [C_i(s_t, a_t, s_{t+1})] \leq w_i\} \tag{5}$$

where $w_i \in \mathbb{R}$. Then the objective for SCMDP is to find a feasible stationary policy from $\bar{\Pi}_C$ that maximizes the performance measure. Formally,

$$\max_{\pi} \ \mathcal{J}_0(\pi), \ \text{s.t.} \ \pi \in \bar{\Pi}_C \tag{6}$$

### 3.3 Maximum Markov Decision Process

Note that for (6), every state-action transition pair corresponds to a constraint, which is intractable to solve using conventional reinforcement learning algorithms. Our intuition is that, instead of directly constraining the cost of each possible state-action transition, we can constrain the expected maximum state-wise cost along the trajectory, which is much easier to solve. Following that intuition, we define a novel *Maximum Markov-Decision Process* (MMDP), which further extends CMDP via (i) a set of up-to-now maximum state-wise costs $\boldsymbol{M} \doteq [M_1, M_2, \cdots, M_m]$ where $M_i \in \mathcal{M} \subset \mathbb{R}$, and (ii) a set of *cost increment* functions, $D_1, D_2, \cdots, D_m$, where $D_i : (\mathcal{S}, \mathcal{M}^m) \times \mathcal{A} \times \mathcal{S} \mapsto [0, \mathbb{R}^+]$ maps the augmented state action transition tuple into a non-negative cost increment. We define the augmented state $\hat{s} = (s, \boldsymbol{M}) \in (\mathcal{S}, \mathcal{M}^m) \doteq \hat{\mathcal{S}}$, where $\hat{\mathcal{S}}$ is the augmented state space. Formally,

$$D_i(\hat{s}_t, a_t, \hat{s}_{t+1}) = \max\{C_i(s_t, a_t, s_{t+1}) - M_{it}, 0\}. \tag{7}$$

By setting $D_i(\hat{s}_0, a_0, \hat{s}_1) = C_i(s_0, a_0, s_1)$, we have $M_{it} = \sum_{k=0}^{t-1} D_i(\hat{s}_k, a_k, \hat{s}_{k+1})$ for $t \geq 1$. Hence, we define *expected maximum state-wise cost* (or $D_i$-return) for $\pi$:

$$\mathcal{J}_{D_i}(\pi) = \mathbb{E}_{\tau \sim \pi} \left[ \sum_{t=0}^{H} D_i(\hat{s}_t, a_t, \hat{s}_{t+1}) \right]. \tag{8}$$

Importantly, (8) is the key component of MMDP and differs our work from existing safe RL approaches that are based on CMDP cost measure (2). With (8), (6) can be rewritten as:

$$\max_{\pi} \mathcal{J}(\pi), \ \text{s.t.} \ \forall i, \mathcal{J}_{D_i}(\pi) \leq w_i, \tag{9}$$

where $\mathcal{J}(\pi) = \mathbb{E}_{\tau \sim \pi} \left[ \sum_{t=0}^{H} \gamma^t R(\hat{s}_t, a_t, \hat{s}_{t+1}) \right]$ and $R(\hat{s}, a, \hat{s}') \doteq R(s, a, s')$. With $R(\tau)$ being the discounted return of a trajectory, we define the on-policy value function as $V^\pi(\hat{s}) \doteq \mathbb{E}_{\tau \sim \pi}[R(\tau)|\hat{s}_0 = \hat{s}]$, the on-policy action-value function as $Q^\pi(\hat{s}, a) \doteq \mathbb{E}_{\tau \sim \pi}[R(\tau)|\hat{s}_0 = \hat{s}, a_0 = a]$, and the advantage function as $A^\pi(\hat{s}, a) \doteq Q^\pi(\hat{s}, a) - V^\pi(\hat{s})$. Lastly, we define on-policy value functions, action-value functions, and advantage functions for the cost increments in analogy to $V^\pi, Q^\pi$, and $A^\pi$, with $D_i$ replacing $R$, respectively. We denote those by $V_{D_i}^\pi, Q_{D_i}^\pi$ and $A_{D_i}^\pi$.

## 4  State-wise Constrained Policy Optimization

To solve large and continuous MDPs, policy search algorithms search for the optimal policy within a set $\Pi_\theta \subset \Pi$ of parametrized policies. In local policy search [Peters and Schaal, 2008], the policy is iteratively updated by maximizing $\mathcal{J}(\pi)$ over a local neighborhood of the most recent policy $\pi_k$. In local policy search for SCMDPs, policy iterates must be feasible, so optimization is over $\Pi_\theta \bigcap \bar{\Pi}_C$. The optimization problem is:

$$\pi_{k+1} = \underset{\pi \in \Pi_\theta}{\text{argmax}} \ \mathcal{J}(\pi), \tag{10}$$
$$\text{s.t.} \ \mathcal{D}ist(\pi, \pi_k) \leq \delta,$$
$$\mathcal{J}_{D_i}(\pi) \leq w_i, i = 1, \cdots, m.$$

where $\mathcal{D}ist$ is some distance measure, and $\delta > 0$ is a step size. For actual implementation, we need to evaluate the constraints first in order to determine the feasible set. However, it is challenging to evaluate the constraints using samples during the learning process. In this work, we propose SCPO inspired by recent trust region optimization methods Schulman et al. [2015]. SCPO approximates (10) using (i) KL divergence distance metric $\mathcal{D}ist$ and (ii) surrogate functions for the objective and constraints, which can be easily estimated from samples on $\pi_k$. Mathematically, SCPO requires the policy update at each iteration is bounded within a trust region, and updates policy via solving following optimization:

$$\pi_{k+1} = \underset{\pi \in \Pi_\theta}{\mathbf{argmax}} \; \underset{\substack{\hat{s} \sim d^{\pi_k} \\ a \sim \pi}}{\mathbb{E}} [A^{\pi_k}(\hat{s}, a)] \tag{11}$$

$$\textbf{s.t.} \;\; \underset{\hat{s} \sim \bar{d}^{\pi_k}}{\mathbb{E}}[\mathcal{D}_{KL}(\pi \| \pi_k)[\hat{s}]] \leq \delta,$$

$$\mathcal{J}_{D_i}(\pi_k) + \underset{\substack{\hat{s} \sim \bar{d}^{\pi_k} \\ a \sim \pi}}{\mathbb{E}}\left[ A^{\pi_k}_{D_i}(\hat{s}, a) \right] + 2(H+1)\epsilon^{\pi}_{D_i}\sqrt{\frac{1}{2}\delta} \leq w_i, i = 1, \cdots, m.$$

where $\mathcal{D}_{KL}(\pi' \| \pi)[\hat{s}]$ is KL divergence between two policy $(\pi', \pi)$ at state $\hat{s}$, the set $\{\pi \in \Pi_\theta : \mathbb{E}_{\hat{s} \sim \bar{d}^{\pi_k}}[\mathcal{D}_{KL}(\pi \| \pi_k)[\hat{s}]] \leq \delta\}$ is called *trust region*, $d^{\pi_k} \doteq (1-\gamma)\sum_{t=0}^{H}\gamma^t P(\hat{s}_t = \hat{s}|\pi_k)$, $\bar{d}^{\pi_k} \doteq \sum_{t=0}^{H} P(\hat{s}_t = \hat{s}|\pi_k)$ and $\epsilon^{\pi}_{D_i} \doteq \max_{\hat{s}}|\mathbb{E}_{a \sim \pi}[A^{\pi_k}_{D_i}(\hat{s}, a)]|$. We then show that SCPO guarantees (i) worst case maximum state-wise cost violation, and (ii) worst case performance degradation for policy update, by establishing new bounds on the difference in returns between two stochastic policies $\pi$ and $\pi'$ for MMDPs.

**Theoretical Guarantees for SCPO** We start with the theoretical foundation for our approach, i.e. a new bound on the difference in state-wise maximum cost between two arbitrary policies. The following theorem connects the difference in maximum state-wise cost between two arbitrary policies to the total variation divergence between them. Here total variation divergence between discrete probability distributions $p, q$ is defined as $\mathcal{D}_{TV}(p\|q) = \frac{1}{2}\sum_i |p_i - q_i|$. This measure can be easily extended to continuous states and actions by replacing the sums with integrals. Thus, the total variation divergence between two policy $(\pi', \pi)$ at state $\hat{s}$ is defined as: $\mathcal{D}_{TV}(\pi'\|\pi)[\hat{s}] = \mathcal{D}_{TV}(\pi'(\cdot|\hat{s})\|\pi(\cdot|\hat{s}))$.

**Theorem 1** (Trust Region Update State-wise Maximum Cost Bound). *For any policies $\pi', \pi$, with $\epsilon^{\pi'}_D \doteq \max_{\hat{s}}|\mathbb{E}_{a \sim \pi'}[A^{\pi}_D(\hat{s}, a)]|$, and define $\bar{d}^\pi = \sum_{t=0}^{H} P(\hat{s}_t = \hat{s}|\pi)$ as the non-discounted augmented state distribution using $\pi$, then the following bound holds:*

$$\mathcal{J}_D(\pi') - \mathcal{J}_D(\pi) \leq \underset{\substack{\hat{s} \sim \bar{d}^\pi \\ a \sim \pi'}}{\mathbb{E}}\left[ A^{\pi}_D(\hat{s}, a) + 2(H+1)\epsilon^{\pi'}_D \mathcal{D}_{TV}(\pi'\|\pi)[\hat{s}] \right]. \tag{12}$$

The proof for Theorem 1 is summarized in Appendix A. Next, we note the following relationship between the total variation divergence and the KL divergence [Boyd et al., 2003, Achiam et al., 2017a]: $\mathbb{E}_{\hat{s} \sim \bar{d}^\pi}[\mathcal{D}_{TV}(p\|q)[\hat{s}]] \leq \sqrt{\frac{1}{2}\mathbb{E}_{\hat{s} \sim \bar{d}^\pi}[\mathcal{D}_{KL}(p\|q)[\hat{s}]]}$. The following bound then follows directly from Theorem 1:

$$\mathcal{J}_D(\pi') \leq \mathcal{J}_D(\pi) + \underset{\substack{\hat{s} \sim \bar{d}^\pi \\ a \sim \pi'}}{\mathbb{E}}\left[ A^{\pi}_D(\hat{s}, a) + 2(H+1)\epsilon^{\pi'}_D\sqrt{\frac{1}{2}\mathbb{E}_{\hat{s} \sim \bar{d}^\pi}[\mathcal{D}_{KL}(\pi'\|\pi)[\hat{s}]]} \right]. \tag{13}$$

By Equation (13), we have a guarantee for satisfaction of maximum state-wise constraints:

**Proposition 1** (SCPO Update Constraint Satisfaction). *Suppose $\pi_k, \pi_{k+1}$ are related by (11), then $D_i$-return for $\pi_{k+1}$ satisfies*

$$\forall i, \mathcal{J}_{D_i}(\pi_{k+1}) \leq w_i.$$

Proposition 1 presents the first constraint satisfaction guarantee under MMDP. Unlike trust region methods such as CPO and TRPO, which assume a discounted sum characteristic, MMDP's non-discounted sum characteristic invalidates these theories. As the maximum state-wise cost is calculated

through a summation of non-discounted increments, analysis must be performed on a finite horizon to upper bound the worst-case summation. In contrast, the theory behind CPO relies on infinite horizon analysis with discounted constraint assumptions, which is not applicable for MMDP settings.

Next, we provide the performance guarantee of SCPO. Previous analyses of performance guarantees have focused on infinite-horizon MDP. We generalize the analysis to finite-horizon MDP, inspired by previous work [Kakade and Langford, 2002, Schulman et al., 2015, Achiam et al., 2017a], and prove it in Appendix B. The infinite-horizon case can be viewed as a special case of the finite-horizon setting.

**Proposition 2** (SCPO Update Worst Performance Degradation). *Suppose $\pi_k, \pi_{k+1}$ are related by* (11), *with $\epsilon^{\pi_{k+1}} \doteq \max_{\hat{s}} |\mathbb{E}_{a \sim \pi_{k+1}}[A^{\pi_k}(\hat{s}, a)]|$, then performance return for $\pi_{k+1}$ satisfies*

$$\mathcal{J}(\pi_{k+1}) - \mathcal{J}(\pi_k) \geq -\frac{\sqrt{2\delta}\gamma\epsilon^{\pi_{k+1}}}{1 - \gamma}.$$

## 5  Practical Implementation

In this section, we show how to (a) implement an efficient approximation to the update (11), (b) encourage learning even when (11) becomes infeasible, and (c) handle the difficulty of fitting augmented value $V_{D_i}^\pi$ which is unique to our novel MMDP formulation. The full SCPO pseudocode is given as algorithm 1 in appendix C.

**Practical implementation with sample-based estimation**   We first estimate the objective and constraints in (11) using samples. Note that we can replace the expected advantage on rewards using an importance sampling estimator with a sampling distribution $\pi_k$ [Achiam et al., 2017a] as

$$\mathbb{E}_{\hat{s} \sim d^{\pi_k}, \, a \sim \pi}[A^{\pi_k}(\hat{s}, a)] = \mathbb{E}_{\hat{s} \sim d^{\pi_k}, \, a \sim \pi_k}\left[\frac{\pi(a|\hat{s})}{\pi_k(a|\hat{s})}A^{\pi_k}(\hat{s}, a)\right]. \tag{14}$$

(14) allows us to replace $A^{\pi_k}$ with empirical estimates at each state-action pair $(\hat{s}, a)$ from rollouts by the previous policy $\pi_k$. The empirical estimate of reward advantage is given by $R(\hat{s}, a, \hat{s}') + \gamma V^{\pi_k}(\hat{s}') - V^{\pi_k}(\hat{s})$. $V^{\pi_k}(\hat{s})$ can be computed at each augmented state by taking the discounted future return. The same can be applied to the expected advantage with respect to cost increments, with the sample estimates given by $D_i(\hat{s}, a, \hat{s}') + V_{D_i}^{\pi_k}(\hat{s}') - V_{D_i}^{\pi_k}(\hat{s})$. $V_{D_i}^{\pi_k}(\hat{s})$ is computed by taking the non-discounted future $D_i$-return. To proceed, we convexify (11) by approximating the objective and cost constraint via first-order expansions, and the trust region constraint via second-order expansions. Then, (11) can be efficiently solved using duality [Achiam et al., 2017a].

**Infeasible constraints**   An update to $\theta$ is computed every time (11) is solved. However, due to approximation errors, sometimes (11) can become infeasible. In that case, we follow [Achiam et al., 2017a] to propose an recovery update that only decreases the constraint value within the trust region. In addition, approximation errors can also cause the proposed policy update (either feasible or recovery) to violate the original constraints in (11). Hence, each policy update is followed by a backtracking line search to ensure constraint satisfaction. If all these fails, we relax the search condition by also accepting decreasing expected advantage with respect to the costs, when the cost constraints are already violated. Denoting $c_i \doteq \mathcal{J}_{D_i}(\pi_k) + 2(H+1)\epsilon_D^\pi \sqrt{\delta/2} - w_i$, the above criteria can be summarized as

$$\mathbb{E}_{\hat{s} \sim \bar{d}^{\pi_k}}[\mathcal{D}_{KL}(\pi \| \pi_k)[\hat{s}]] \leq \delta \tag{15}$$

$$\mathbb{E}_{\hat{s} \sim \bar{d}^{\pi_k}, a \sim \pi}\left[A_{D_i}^{\pi_k}(\hat{s}, a)\right] - \mathbb{E}_{\hat{s} \sim \bar{d}^{\pi_k}, a \sim \pi_k}\left[A_{D_i}^{\pi_k}(\hat{s}, a)\right] \leq \max(-c_i, 0). \tag{16}$$

Note that the previous expected advantage $\mathbb{E}_{\hat{s} \sim \bar{d}^{\pi_k}, a \sim \pi_k}\left[A_{D_i}^{\pi_k}(\hat{s}, a)\right]$ is also estimated from rollouts by $\pi_k$ and converges to zero asymptotically, which recovers the original cost constraints in (11).

**Imbalanced cost value targets**   A critical step in solving (11) is to fit the cost increment value functions $V_{D_i}^{\pi_k}(\hat{s}_t)$. By definition, $V_{D_i}^{\pi_k}(\hat{s}_t)$ is equal to the maximum cost increment in any future state over the maximal state-wise cost so far. In other words, the true $V_{D_i}^{\pi_k}$ will always be zero for all $\hat{s}_{t:H}$ when the maximal state-wise cost has already occurred before time $t$. In practice, this causes the distribution of cost increment value function to be highly zero-skewed and makes the fitting very hard. To mitigate the problem, we sub-sample the zero-valued targets to match the population of non-zero values. We provide more analysis on this trick in Q3 in section 6.2.

## 6  Experiments

In our experiments, we aim to answer these questions:

**Q1** How does SCPO compare with other state-of-the-art methods for safe RL?

**Q2** What benefits are demonstrated by constraining the maximum state-wise cost?

**Q3** How do the sub-sampling trick of SCPO impact its performance?

### 6.1  Experiment Setups

**New Safety Gym**   To showcase the effectiveness of our state-wise constrained policy optimization approach, we enhance the widely recognized safe reinforcement learning benchmark environment, Safety Gym Ray et al. [2019], by incorporating additional robots and constraints. Subsequently, we perform a series of experiments on this augmented environment.

Our experiments are based on five different robots: (i) **Point:** Figure 2a A point-mass robot ($\mathcal{A} \subseteq \mathbb{R}^2$) that can move on the ground. (ii) **Swimmer:**  Figure 2b A three-link robot ($\mathcal{A} \subseteq \mathbb{R}^2$) that can move on the ground. (iii) **Walker:**  Figure 2d A bipedal robot ($\mathcal{A} \subseteq \mathbb{R}^{10}$) that can move on the ground. (iv) **Ant:**  Figure 2c A quadrupedal robot ($\mathcal{A} \subseteq \mathbb{R}^8$) that can move on the ground. (v) **Drone:** Figure 2e A quadrotor robot ($\mathcal{A} \subseteq \mathbb{R}^4$) that can move in the air.

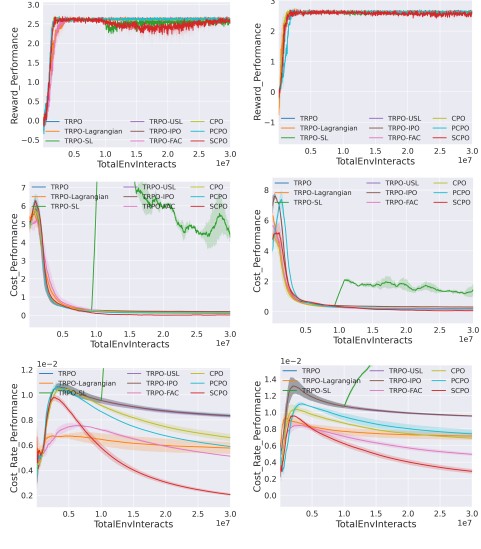

(a) Ant-Hazard-8        (b) Walker-Hazard-8

Figure 1: Comparison of results from two representative test suites in high dimensional systems (Ant and Walker).

All of the experiments are based on the goal task where the robot must navigate to a goal. Additionally, since we are interested in episodic tasks (finite-horizon MDP), the environment will be reset once the goal is reached. For the robots that can move in 3D spaces (e.g, the Drone robot), we also design a new 3D goal task with a sphere goal floating in the 3D space. Three different types of constraints are considered: (i) **Hazard**: Dangerous areas as shown in Figure 3a. Hazards are trespassable circles on the ground. The agent is penalized for entering them. (ii) **3D Hazard**: 3D Dangerous areas as shown in Figure 3b. 3D Hazards are trespassable spheres in the air. The agent is penalized for entering them. (iii) **Pillar**: Fixed obstacles as shown in  Figure 3c. The agent is penalized for hitting them.

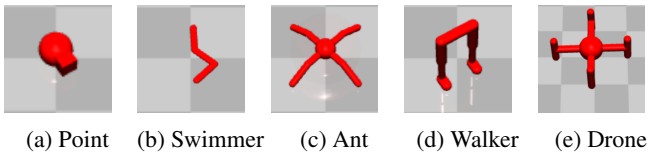

(a) Point    (b) Swimmer    (c) Ant    (d) Walker    (e) Drone

Figure 2: Robots for benchmark problems in upgraded Safety Gym.

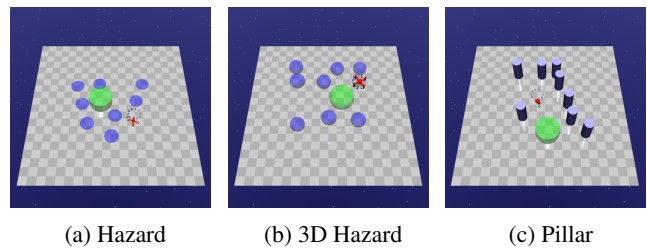

(a) Hazard        (b) 3D Hazard        (c) Pillar

Figure 3: Constraints for benchmark problems in upgraded Safety Gym.

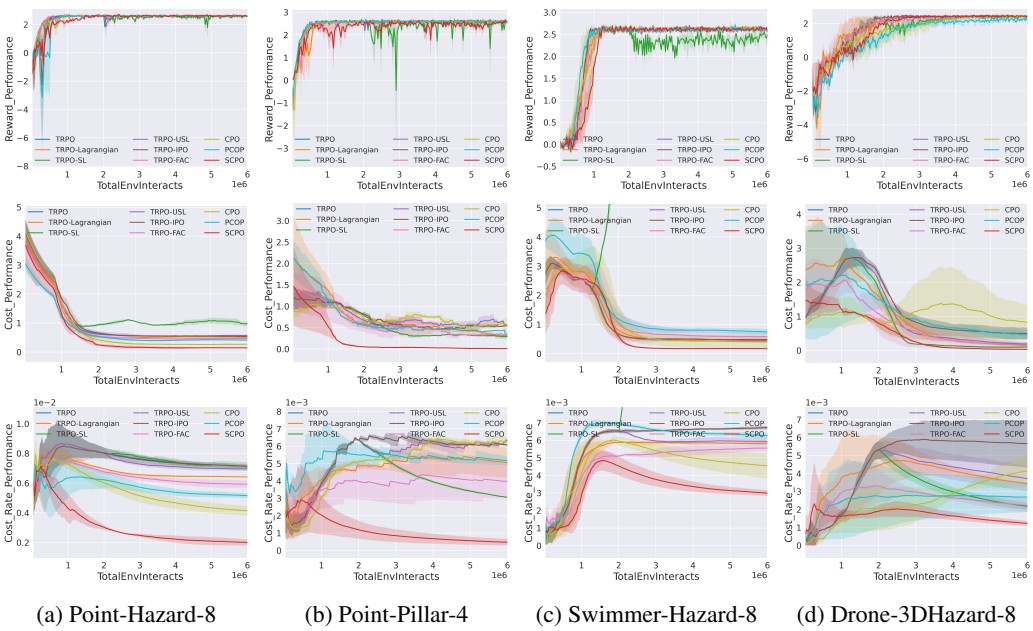

(a) Point-Hazard-8     (b) Point-Pillar-4     (c) Swimmer-Hazard-8     (d) Drone-3DHazard-8

Figure 4: Comparison of results from four representative test suites in low dimensional systems (Point, Swimmer, and Drone).

Considering different robots, constraint types, and constraint difficulty levels, we design 14 test suites with 5 types of robots and 9 types of constraints, which are summarized in Table 1 in Appendix. We name these test suites as `{Robot}-{Constraint Type}-{Constraint Number}`.

**Comparison Group** The methods in the comparison group include: (i) unconstrained RL algorithm TRPO [Schulman et al., 2015] (ii) end-to-end constrained safe RL algorithms CPO [Achiam et al., 2017a], TRPO-Lagrangian [Bohez et al., 2019], TRPO-FAC [Ma et al., 2021], TRPO-IPO [Liu et al., 2020], PCPO [Yang et al., 2020b], and (iii) hierarchical safe RL algorithms TRPO-SL (TRPO-Safety Layer) [Dalal et al., 2018], TRPO-USL (TRPO-Unrolling Safety Layer) [Zhang et al., 2022b]. We select TRPO as our baseline method since it is state-of-the-art and already has safety-constrained derivatives that can be tested off-the-shelf. For hierarchical safe RL algorithms, we employ a warm-up phase ($1/3$ of the whole epochs) which does unconstrained TRPO training, and the generated data will be used to pre-train the safety critic for future epochs. For all experiments, the policy $\pi$, the value $(V^\pi, V_D^\pi)$ are all encoded in feedforward neural networks using two hidden layers of size (64,64) with tanh activations. More details are provided in Appendix D.

**Evaluation Metrics** For comparison, we evaluate algorithm performance based on (i) reward performance, (ii) average episode cost and (iii) cost rate. Comparison metric details are provided in Appendix D.3. We set the limit of cost to 0 for all the safe RL algorithms since we aim to avoid any violation of the constraints. For our comparison, we implement the baseline safe RL algorithms exactly following the policy update / action correction procedure from the original papers. We emphasize that in order for the comparison to be fair, we give baseline safe RL algorithms every advantage that is given to SCPO, including equivalent trust region policy updates.

## 6.2 Evaluating SCPO and Comparison Analysis

**Low Dimension System** We select four representative test suites on low dimensional system (Point, Swimmer, Drone) and summarize the comparison results on Figure 4, which demonstrate that SCPO is successful at approximately enforcing zero constraints violation safety performance in all environments after the policy converges. Specifically, compared with the baseline safe RL methods, SCPO is able to achieve (i) near zero average episode cost and (ii) significantly lower cost rate without sacrificing reward performance. The baseline end-to-end safe RL methods (TRPO-Lagrangian, TRPO-FAC, TRPO-IPO, CPO, PCPO) fail to achieve the near zero cost performance

even when the cost limit is set to be 0. The baseline hierarchical safe RL methods (TRPO-SL, TRPO-USL) also fail to achieve near zero cost performance even with an explicit safety layer to correct the unsafe action at every time step. End-to-end safe RL algorithms fail since all methods rely on CMDP to minimize the discounted cumulative cost while SCPO directly work with MMDP to restrict the state-wise maximum cost by Proposition 1. We also observe that TRPO-SL fails to lower the violation during training, due to the fact that the linear approximation of cost function $C(\hat{s}_t, a, \hat{s}_{t+1})$ [Dalal et al., 2018] becomes inaccurate when the dynamics are highly nonlinear like the ones we used in MuJoCo [Todorov et al., 2012]. More detailed metrics for comparison and experimental results on test suites with low dimension systems are summarized in Appendix D.3.

**High Dimension System**    To demonstrate the scalability and performance of SCPO in high-dimensional systems, we conducted additional tests on the Ant-Hazard-8 and Walker-Hazard-8 suites, with 8-dimensional and 10-dimensional control spaces, respectively. The comparison results for high-dimensional systems are summarized in Figure 1, which show that SCPO outperforms all other baselines in enforcing zero safety violation without compromising performance in terms of return. SCPO rapidly stabilizes the cost return around zero and significantly reduces the cost rate, while the other baselines fail to converge to a policy with near-zero cost. The comparison results of both low dimension and high dimension systems answer **Q1**.

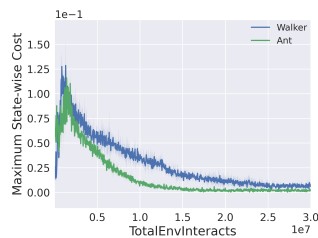

Figure 5:
Maximum state-wise cost

**Maximum State-wise Cost**    As pointed in Section 3.3, the underlying magic for enabling near-zero safety violation is to restrict the maximum state-wise cost to stay around zero. To have a better understanding of this process, we visualize the evolution of maximum state-wise cost for SCPO on the challenging high-dimensional Ant-Hazard-8 and Walker-Hazard-8 test suites in Figure 5 , which answers **Q2**.

**Ablation on Sub-sampling Imbalanced Cost Increment Value Targets**    As pointed in Section 5, fitting $V_{D_i}^{\pi_k}(\hat{s}_t)$ is a critical step towards solving SCPO, which is challenging due to zero-skewed distribution of cost increment value function. To demonstrate the necessity of sub-sampling for solving this challenge, we compare the performance of SCPO with and without sub-sampling trick on the aerial robot test suite, summarized in Figure 6. It is evident that with sub-sampling, the agent achieves higher rewards and more importantly, converges to near-zero costs. That is because sub-sampling effectively balances the cost increment value targets and improves the fitting of $V_{D_i}^{\pi_k}(\hat{s}_t)$. We also attempted to solve the imbalance issue via over-sampling non-zero targets, but did not observe promising results. This ablation study provides insights into **Q3**.

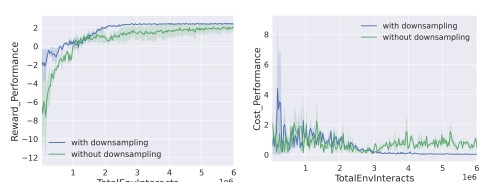

Figure 6: SCPO sub-sampling ablation study with Drone-3DHazard-8

## 7    Conclusion and Future Work

This paper proposed SCPO, the first general-purpose policy search algorithm for state-wise constrained RL. Our approach provides guarantees for state-wise constraint satisfaction at each iteration, allows training of high-dimensional neural network policies while ensuring policy behavior, and is based on a new theoretical result on Maximum Markov Decision Process. We demonstrate SCPO's effectiveness on robot locomotion tasks, showing its significant performance improvement compared to existing methods and ability to handle state-wise constraints.

**Limitation and future work**    One limitation of our work is that, although SCPO satisfies state-wise constraints, the theoretical results are valid only in expectation, meaning that constraint violations are still possible during deployment. To address that, we will study absolute state-wise constraint satisfaction, i.e. bounding the *maximal possible* state-wise cost, which is even stronger than the current result (satisfaction in expectation).

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
