# OpenReview forum: "State-wise Constrained Policy Optimization"
_NeurIPS.cc/2023/Conference — Submitted to NeurIPS 2023_

### Official Review · Reviewer_2Qxj · 2023-07-05

**Soundness:** 2 fair
**Presentation:** 3 good
**Contribution:** 2 fair
**Rating:** 4
**Confidence:** 3

**Summary:**

This paper discusses an important topic about safe reinforcement learning, which explores the state-wise issue. It is a significant problem because, in the real world, state-wise constraints are one of the most common and challenging constraints in safety-critical applications. Most safe RL methods focus on cumulative safety, which may need to be improved to ensure safety during deploying RL in real-robot applications.

**Strengths:**

1. The writing quality is good.
2. Theoretical results sound good to me.
3. The code is provided in this study.


**Weaknesses:**

1. The experimental results are very weird, especially for the reward performance. For example, in Figure 1, all the baselines’ reward values are almost the same; please check the environments' reward settings. This is also shown in Figure 4.
2. The method cannot ensure safety while deploying the method in real-world applications; demos that the authors provided also confirm this point, that is, sometimes the agent will violate safety constraints and crash into obstacles.


**Questions:**

1. How does the study consider high-dimensional optimization? Generally, RL algorithms can handle 8- or 10-dimensional tasks on MuJoCo tasks, e.g., CPO, PPO Lagrangian, and CRPO.
2. How does the study define the reward and cost settings?
3. Why does this study categorize CSC and safety Layer methods as hierarchical policy? Does this study mean these methods are HRL?


**Limitations:**

1. It is hard for me to find some new insights from this study; the problem and the solution are not new.  As for the problem, it is already defined in many papers, e.g., CPO, PCPO, and CRPO; as for the solution, it just revises the cost setting from a trajectory, and does not ensure safety at each step.
2. As for the theoretical analysis, most of the findings could be known from CPO by revising the cost optimization settings.

---

> ### Author Rebuttal · Authors · 2023-08-08
>
> **W1**: In continuous control challenges, such as robot locomotion tasks, performance convergence tends to cluster around a similar magnitude, as rewards have constrained upper bounds due to episodic nature. As long as the goal can be achieved within an episode, performance remains consistent across algorithms. This assertion aligns with findings in established safe reinforcement learning literature, including works like FAC and AutoCost, etc.
>
> A crucial aspect we wish to emphasize is the varying rate of reward convergence among different algorithms, illustrated in Fig 4(c) and (d). The reason for the resemblance between Fig 1 across algorithms lies in the extensive training duration of 3e7 steps, where all algorithms have reached convergence. However, when training is limited to 6e6 steps, Fig 4(c) and (d) underscore significant performance disparities.
>
> In light of these considerations, we contend that the observed performance patterns in Fig 1 and Fig 4 are reasonable and representative of the context.
>
>
> **W2**: While there are indeed safe RL methodologies capable of ensuring zero safety violations, they often come at the cost of demanding extensive prior knowledge regarding system dynamics. This has been highlighted in the recent survey, titled 'State-wise Safe Reinforcement Learning: A Survey,' appeared in the IJCAI Survey Track of 2023.
> We acknowledge that our work ensures state-wise safety constraint satisfaction on an expected level. Nonetheless, this represents a significant stride towards real-world application of safe RL. Existing endeavors primarily handle cumulative discounted cost sums, whereas we have pioneered the direct handling of state-wise safety constraints devoid of any presumptions about system dynamics.
>
> Additionally, it's essential to recognize that the Markov Decision Process framework inherently assigns a non-zero probability to each state visitation. In essence, any safe RL solution for MDP will invariably account for the statistical possibility of encountering unsafe states.
>
> **Q1**: Our work constitutes a constrained policy optimization algorithm, designed to be agnostic of the dimensionality of the action space. This implies that our algorithm is equipped to accommodate action spaces of arbitrary high dimensions, akin to other policy optimization-based safe RL approaches.
>
> Indeed, we have substantiated the capabilities of our algorithm in handling tasks with elevated dimensions, as evidenced by our experimentation (Figure 1) with Mujoco environments such as Ant locomotion (8 dimensions) and Walker locomotion (10 dimensions). These action spaces are detailed from line 263 to line 267, while the corresponding observation spaces are concisely summarized in Table 2 of the appendix.
>
> **Q2**: We have clarified the cost and reward settings in the appendix: "E Experiment Details". And "E.1 Environment Settings/ Goal Task" defines the reward function which is the reward settings. "E.1 Environment Settings/ Hazard Constraint" defines the cost function which is the cost settings.
>
> **Q3**:  The reason for this categorization is that both CSC and safety layer methods incorporate a hierarchical structure in their policy design. The higher-level policy outputs reference actions, which are then subject to correction or projection by the safety monitor at the lower level.
> However, this categorization should not be mistaken for Hierarchical Reinforcement Learning (HRL), which pertains to the decomposition of long-horizon decision-making processes into simpler sub-tasks to facilitate more efficient learning.
>
> The categorization is also consistent with the recently published survey paper, "State-wise Safe Reinforcement Learning: A Survey," appeared at IJCAI Survey Track 2023. We will ensure that this clarification is duly highlighted in the revised manuscript.
>
>
> **L1**: We acknowledge that CMDPs have been well-studied in the literature, with many safe RL methods proposed, such as CPO, PCPO, CRPO, Lagrangian and more. These approaches indeed tackled bounding cumulative cost in CMDPs. However, our problem is fundamentally different.
>
> Our work focuses on the problem of SCMDPs, where safety constraints are instantaneous, rather than cumulative. This distinction has a profound impact on the problem's complexity and the applicability of existing solutions. The SCMDP formulation invalidates all previous approaches and theoretical results that were designed specifically for CMDPs.
>
> Moving to the solution, our method is the first to provide expected state-wise safety satisfaction guarantee in the framework of policy optimization.
>
> In conclusion, we respectfully assert that our work addresses a significantly different problem from existing studies in safe RL.
>
> **L2**: While it is true that both our work and CPO aim to address safe RL problems, there are fundamental differences.
>
> Firstly, our work explicitly considers the finite horizon MDP, i.e. MMDP. This finite horizon aspect is critical, as it fundamentally alters the problem setup compared to the infinite horizon MDPs typically considered in CPO. It is worth noting that the transformation matrix of a finite horizon MDP is non-stationary, which indicates the finite horizon is not a special case of the infinite horizon problem. The finite horizon nature of MMDP significantly impacts the mathematical derivations and introduces new challenges in theoretical analysis.
>
> Secondly, the summation is non-discounted in MMDP, whereas CPO and other related works rely on discounted sum over infinite horizon trajectories. This difference in the summation type directly invalidates the proofs from CPO and renders them inapplicable to the MMDP framework.
>
> Consequently, any theoretical result obtained in MMDP requires a separate, dedicated derivation process. As such, our work pioneers the exploration of trust region-based techniques in the context of non-discounted MDPs and finite horizon settings.

---

### Official Review · Reviewer_teqG · 2023-07-07

**Soundness:** 3 good
**Presentation:** 3 good
**Contribution:** 2 fair
**Rating:** 4
**Confidence:** 3

**Summary:**

This work tackles the problem of state-wise safety in the reinforcement learning problem. To this end it introduces the framework of Maximum Markov Decision Process and an algorithm State-wise Constrained Policy Optimization (SCPO) to solve the problem. Numerical results illustrate the performance of the authors algorithm.

**Strengths:**

The paper is well written and supported by extensive numerical results.


**Weaknesses:**

Novelty is slightly limited. Similar state augmentation mechanisms have been studied before. The algorithm developed here arises from a somewhat straightforward application of trust region methods to such state augmentation mechanisms.


**Questions:**

Additional details on the motivation behind the Maximum Markov Decision Process formulation would be beneficial. At first read it is slightly confusing as to why state-wise constraints are introduced under this formulation.

The framework of Maximum Markov Decision Process introduces a state augmentation. These mechanisms have been used in the past [Sootla, Calvo-Fullana]. Specially as the augmentation introduced by the authors is similar to augmenting the state space with the dual variables. While the algorithm introduced by the authors here is clearly different, a comparison with these augmentation methods should be done.

A. Sootla, A. I. Cowen-Rivers, J. Wang, and H. B. Ammar. Enhancing safe exploration using safety state augmentation. arXiv preprint arXiv:2206.02675, 2022
M. Calvo-Fullana, S. Paternain, L. F. O. Chamon, and A. Ribeiro. State augmented constrained reinforcement learning: Overcoming the limitations of learning with rewards. arXiv preprint arXiv:2102.11941, 2021.

**Limitations:**

Addressed.

---

> ### Author Rebuttal · Authors · 2023-08-08
>
> **Q1**:  Thank you for bringing this to our attention! We mentioned from line 141 to line 144 that enforcing constraints on every state-action transition pair in SCMDP, i.e. eq. (5),  is challenging.  As an alternative, we propose constraining the maximum state-wise cost along the trajectory, which allows us to regulate every state-action constraint by controlling the upper limit of costs.
> Therefore, we introduce the concept of Maximum Markov Decision Process (MMDP), so that state-wise constraints are transformed into maximum cost constraints.
>
>
> **Q2**: Thank you for drawing our attention to the state augmentation techniques used in the works of Sootla and Calvo-Fullana [Sootla, Calvo-Fullana]. We appreciate the opportunity to compare SCPO with these previous works in terms of state augmentation. While state augmentation is a common element, it is not the major contribution of SCPO, and there are significant differences between our approach and those in Sootla and Calvo-Fullana's works. Below, we summarize the core ideas of each approach and highlight their correlations and differences.
>
> Core Idea of Calvo-Fullana: Calvo-Fullana's work is built on the Lagrangian method, where a Lagrangian multiplier $\lambda$ plays a crucial role in achieving policy optimality. However, selecting an appropriate $\lambda$ is a challenging task. To overcome this challenge, Calvo-Fullana introduces a dynamic adaptation of $\lambda$ at every time step to capture the degree of constraint satisfaction. Consequently, the policy is trained to maximize rewards with respect to specific $\lambda$, necessitating state augmentation of both the state $s$ and the Lagrangian multiplier $\lambda$.
>
> Core Idea of Sootla: Sootla's primary insight lies in equating the enforcement of safety constraints to ensuring that the remaining safety budget remains larger than zero. Here, the safety budget represents the distance towards constraint violation. With this understanding, the agent can make informed decisions regarding aggressive actions, thus respecting safety constraints. The paper further introduces a curriculum training scheme where the initial remaining safety budget target is gradually increased, starting with a stricter curriculum and progressively expanding to an easier one. This approach aims to reduce constraint violations during training, as the agent learns to handle scenarios with tighter safety budgets effectively.
>
> Core Idea of SCPO: In SCPO, we utilize state augmentation within the context of Maximum Markov Decision Process (MMDP). The state space is augmented with an "up-to-now maximum state-wise cost value," enabling the computation of "maximum state-wise cost increment" at each step in a Markovian manner. As such, maximum state-wise cost can be computed via non-discounted summation of “cost increments”, making it feasible to employ trust region based techniques to provide state-wise safety guarantees.
>
> Correlation: All three works involve augmenting the state space with an additional value to capture partial information related to safety constraint satisfaction.
>
> Difference: The augmentation serves different purposes and functionalities in each work. (1) Calvo-Fullana's Lagrangian multiplier augmentation enables policy optimization with varying multipliers. (2)  Sootla uses remaining safety budget (RSB) augmentation so that a curriculum training can be introduced via solving constrained RL with increasing RSB. It is worth noting RSB is under the concept of cumulative discounted cost sum, and RSB is specified a priori in the curriculum training. (3) SCPO's "up-to-now maximum state-wise cost value" augmentation facilitates the computation of "maximum state-wise cost increment" for MMDP in a Markovian fashion. Here “up-to-now maximum state-wise cost value” is under the scope of instantaneous cost, and is updated as a posterior metric policy training.
>
> In our final version, we will cite these works and include this comparative analysis for readers to understand the differences between SCPO and these state augmentation works .
>
>
> **W1**: Thanks for the feedback. Firstly, different from other state augmentation safe RL methods that work with CDMP, our work directly handles SCMDP with instant safety constraints at each step, and our state augmentation purpose and functionality are fundamentally different as pointed out in the previous answers.
>
> Secondly, our work explicitly considers the finite horizon MDP, i.e. MMDP, where non-discounted summation is required and the state transformation is non-stationary.  This finite horizon non-discounted aspect is critical, as it fundamentally alters the problem setup compared to the infinite horizon discounted summation considered in previous trust region methods. This difference in the summation type directly invalidates the theoretical results from TRPO, CPO, PCPO and other trust region methods and renders them inapplicable to the MMDP framework.
>
> In conclusion, while the state augmentation mechanism forms a component of our approach, the true novelty lies in SCMDP problem formulation, MMDP framework, and state-wise safety guarantees with SCPO. We will make sure to highlight these points in the revised manuscript.

---

### Official Review · Reviewer_aqq3 · 2023-07-24

**Soundness:** 3 good
**Presentation:** 3 good
**Contribution:** 4 excellent
**Rating:** 7
**Confidence:** 3

**Summary:**

This paper introduces a novel approach to solve safe RL tasks. This approach is based on a new method SCPO and corresponding MMDP framework. Authors show the efficacy of this method by providing mathematical guarantees. Additionally, authors give useful practical implementation tips to improve reproducibility of their work.

**Strengths:**

originality
  - This paper extends CMDP framework to account for state-wise safety guarantees. It's a fairly natural extension, and authors did a great job of explaining the shortcomings of existing methods and how theirs addresses the gaps.

quality
  - The paper is well-written and includes thorough mathematical analysis and practical tips for implementation.
  - Experimental section includes fair comparison with existing SOTA methods.

clarity
  - The paper uses consistent notation with previous papers (i.e. TRPO).
  - Including pseudocode (in appendix) is useful.

significance.
  - The paper has a good significant impact on safe RL. The presented method is novel and beats existing methods on the selected benchmarks. Authors also provided concrete future direction for this work.

**Weaknesses:**

- Not necessarily a weakness, but including intuitive explanation of proposition 1 and 2 would be helpful to increase overall readability of the paper.
- Paper heavily relied on concepts TRPO, but didn't give explanations about them in the background section.

**Questions:**

N/A

**Limitations:**

Yes

---

> ### Author Rebuttal · Authors · 2023-08-08
>
> Thank you for your valuable feedback on our paper. We sincerely appreciate your time and effort in reviewing our work. Please find our answers to your questions below.
>
> **Q1**: “Not necessarily a weakness, but including intuitive explanation of proposition 1 and 2 would be helpful to increase overall readability of the paper.”
>
> **A1**: We would like to express our gratitude for your valuable feedback. We fully agree that providing intuitive explanations of Propositions 1 and 2 will significantly enhance the readability and comprehension of our paper.
>
> Proposition 2 establishes a fundamental result that bounds the performance degradation when policy updates are carried out via solving Problem 11, which ensures satisfaction of the trust region step size constraint and the state-wise maximum cost constraints. Intuitively, this proposition assures that when our policy is updated within these specified constraints, the degradation in reward performance will be limited. This means that our approach strikes a balance between improving the policy's performance and satisfying the state-wise safety constraints. Similarly, Proposition 1 showcases another essential finding, demonstrating that by optimizing the policy using Problem 11, we can guarantee satisfaction of the state-wise safety constraints in the State-wise Constrained Markov Decision Process (SCMDP) setting.
>
> In our subsequent version, we will certainly incorporate these intuitive explanations into the paper. Once again, we appreciate your suggestion.
>
> **Q2**: “Paper heavily relied on concepts TRPO, but didn't give explanations about them in the background section.”
>
> **A2**: We sincerely appreciate your comprehensive evaluation and valuable observations. Your feedback has highlighted an important aspect of our paper, and we recognize the significance of providing a thorough background on Trust Region Policy Optimization (TRPO) to aid readers in understanding the foundations and logic of SCPO.
>
> In our revised version, we will ensure to include a detailed explanation of TRPO in the background section. Specifically, we will elaborate on how the trust region method constrains the difference between the new policy and the old policy within a limited Kullback-Leibler (KL) divergence, known as the trust region. The purpose should be to let the reader better understand the difference between the proposed method and TRPO, i.e., differentiate the proposed work from prior work. Once again, we want to express our utmost gratitude for your support and invaluable contributions.

---

### Official Review · Reviewer_DdAh · 2023-07-26

**Soundness:** 3 good
**Presentation:** 3 good
**Contribution:** 3 good
**Rating:** 7
**Confidence:** 4

**Summary:**

This paper introduces a novel framework called Maximum MDP to address the problem of state-wise constrained policy optimization, namely the authors considers limiting the expected maximum state-wise cost rather than the cost for each state. Similar to the TRPO/CPO framework, the authors derived a worse-case constraint violation guarantee and practical algorithm which were shown in be effective in a set of robotic locomotive tasks.

**Strengths:**

- I really like the simplicity of the idea proposed by the authors, state-wise safe RL is a much more common requirement for real-world safety critical problems but most literature on safe RL focus on constraining the cumulative cost. Reformulating the problem as constraining expected maximum state-wise cost eliminates the scalability issues associated with state-wise safe RL and effectively transforms the problem into a cumulative cost problem.
- Furthermore, this formulation allows the authors to easily adapt existing algorithms for cumulative constraints (CPO) with little modifications.
- Experiment section is well-structured and answers key practical questions associated with the proposed algorithms.

**Weaknesses:**

- Theorem 1 doesn't really depend on the specific formulation of MMDP and is really a finite-horizon variant of the policy improvement theorem from TRPO/CPO (which I think is a nice contribution in itself), I think the authors should try to convey this
- There is a major error in the proof in the appendix, note that equation 20 is not correct since $I - P$ is not invertible

**Questions:**

- One related reference which seems to be missing is [1] which similarly uses a trust region based method but for cumulative costs, one question I have here is can SCPO be applied to other trust-region based algorithms such as PCPO or FOCOPS?
- Overall, I do like the main idea proposed by the authors and I think the general approach is sound and the empirical improvements are valid. However the issue in the proof essentially invalidates Theorem 1. While I am fairly sure a finite horizon variant of the policy improvement theorem from CPO/TRPO does exist, the authors need to make sure the results are correct. At present, I cannot recommend this paper for acceptance at this venue, however I am willing to revise my score if the authors do fix the mathematical issues during the rebuttal period.
- [Minor] Achiam et al.'s Constrained Policy Optimization seem to appear twice in your list of references

[1] Zhang et al. (2020) First Order Constrained Optimization in Policy Space

**Limitations:**

Yes

---

> ### Author Rebuttal · Authors · 2023-08-08
>
> We sincerely thank you for your comprehensive comments on our paper and please find our answers to your questions below.
>
> **W1**: We totally agree that our Theorem 1 doesn't really depend on MMDP. Thanks very much for pointing that out! We will definitely add an explanation about this in the latter version of our paper.
>
> **W2**: Thank you for your insightful review of the manuscript! That’s a great catch! We have reworked the proof accordingly as follow:
>
> The overall idea is to first get the upper bound of discounted cumulative cost for finite horizon MDP (discussed in Appendix B), then we show the upper bound of non-discounted sum for finite horizon can be obtained via taking the limits of discounted sum version result.
>
> We make three small updates to the proof of discounted sum results:
> Firstly, In order to prevent the appearance of $\frac{1}{1-\gamma}$ which may leads to the singularity of functions we used, the $d^\pi$ we defined in eq (39) is changed to $\dot d^\pi$:
> \begin{align}
>   & \dot d^\pi(\hat s)=\sum\_{t=0}^H\gamma^t P(\hat s\_t=\hat s|\pi)
> \end{align}
>
> Secondly, eq. (41) is updated as:
> \begin{align}
>   \dot d^\pi& =\sum\limits\_{t=0}^H(\gamma P\_\pi)^t \hat \mu \\\\
>   &=(I-(\gamma P\_\pi)^{H+1})(I-\gamma P\_\pi)^{-1}\hat \mu \\\\
>   &=(I-\gamma P\_\pi)^{-1}\hat \mu
> \end{align}
>
> Noticing that the finite MDP ends up at step $H$, thus $(P\_{\pi})^{H+1}$ should be set to zero matrix.
>
> Thirdly, eq. (55) is updated as:
> \begin{align}
>   \|\bar{G}\|\_1=\|(I-\gamma P\_{\pi'})^{-1}\|\_1\le\sum\_{t=0}^\infty\gamma^t\|P\_{\pi'}^t\|\_1=\sum\_{t=0}^H\gamma^t
> \end{align}
>
> Thus, following the similar derivations as the proof of Theorem 2. The discounted sum result yields following inequality:
> \begin{align}
>  &\mathcal{\hat J}\_D(\pi') - \mathcal{\hat J}\_D(\pi) \leq \underset{\substack{\hat s\sim \dot d^\pi, a\sim\pi'}}{E}\left[\hat A\_D^\pi(\hat s,a) + 2(\sum\_{t=0}^H\gamma^{t+1})\hat \epsilon\_{D}^{\pi'} \mathcal{D}\_{TV}(\pi'||\pi)[\hat s]\right]
> \end{align}
>
> where $\mathcal{\hat J}\_D(\pi) = \mathbb{E}\_{\tau \thicksim \pi} \Bigg[ \sum\_{t=0}^{H} \gamma^t D\big(\hat s\_t, a\_t, \hat s\_{t+1}\big)\Bigg]$, need to distinguish from $\mathcal{J}\_D(\pi)$. And $\hat V\_D^{\pi}, \hat A\_D^\pi, \hat \epsilon\_{D}^{\pi'}$ are also the discounted version of $V\_D^{\pi}$,  $A\_D^\pi, \epsilon\_{D}^{\pi'}$. Note that one can only get this the inequality holds when $\gamma \in (0,1)$.
>
> Then we can define $\mathcal{F}(\gamma) = \underset{\substack{\hat s\sim \dot d^\pi, a\sim\pi'}}{E}\left[\hat A\_D^\pi(\hat s,a) + 2(\sum\_{t=0}^H\gamma^{t+1})\hat\epsilon\_{D}^{\pi'} \mathcal{D}\_{TV}(\pi'||\pi)[\hat s]\right] - \mathcal{\hat J}\_D(\pi') + \mathcal{\hat J}\_D(\pi)$. Since the expectation terms of $\mathcal{F}(\gamma)$ are taken over polynomials of $\gamma$, and the expectations will not work on $\gamma$ directly, we can infer that $\mathcal{F}(\gamma)$ is a polynomial function with respect to $\gamma$. Therefore, the following conditions hold:
> \begin{align}
>   &\mathcal{F}(\gamma) \geq 0, \text{when}~ \gamma \in (0,1)\\\\
>   &\mathcal{F}(\gamma)\text{ is a polynomial function}
> \end{align}
>
> Because $\mathcal{F}(\gamma)$ is a polynomial function with bounded coefficients, $\underset{\gamma\rightarrow 1^-}{\lim}\mathcal{F}(\gamma)$ exists and $\mathcal{F}(\gamma)$ is continuous at point $(1, \mathcal{F}(1)$). Thus, $\mathcal{F}(1) = \underset{\gamma\rightarrow 1^-}{\lim}\mathcal{F}(\gamma)\geq 0$, which indicates:
> \begin{align}
>  &\mathcal{J}\_D(\pi') - \mathcal{J}\_D(\pi) \leq \underset{\substack{\hat s\sim \bar d^\pi, a\sim\pi'}}{E}\left[A\_D^\pi(\hat s,a) + 2(H+1)\epsilon\_{D}^{\pi'} \mathcal{D}\_{TV}(\pi'||\pi)[\hat s]\right].
> \end{align}
>
> where $\bar d^{\pi} = \sum\_{t=0}^H P(\hat s\_t=\hat s|\pi)$.
>
>
>
> **Q1**: Thanks for the suggestion! We will add this reference in the revised version. As for the feasibility of extending SCPO to other trust region based algorithms, the answer is YES:
>
> PCPO: In order to solve the problem about the infeasibility of finding the solution in CPO, PCPO put the maximization of reward in the first place and then projected the policy into a feasible set defined by CMDP. In SCPO, we can also maximize the reward first and project the policy into feasible policy set wrt. SCMDP. There are no theoretical difficulties to stop us from doing this.
>
> FOCOPS: The situation is the same for the FOCOPS method. It solves the constrained policy optimization problem by two steps: Firstly, it solves the problem in the non parameterized policy space. And then it projects the policy found in the previous step back into the parameterized policy space by solving for the closest policy. The two steps could work for SCPO as well because the constrained policy optimization problem is also solvable in the non parameterized policy space. The projection step works as well.
>
> **Q2**: Thank you for your recognition and appreciation! As for the new proof, please see the proof above for details.
>
> **Q3**: Good catch! We'll fix it in the latter version.

---

> > ### Comment · Reviewer_DdAh · 2023-08-20
> > **Response to author rebuttal**
> >
> > I want to thank the authors for you detailed rebuttal. I have looked through the authors rebuttal as well as reviews from other reviewers. I believe the proof for Theorem 1 is now correct, though just one quick comment is that intuitively I think a simpler proof may exist by simply taking advantage of the equivalence between infinite horizon discounted and finite horizon undiscounted MDPs (see Chapter 5 of Puterman), but there could be complications I have not considered.
> >
> > Since my main concern with the original paper was the mathematical proof, I am satisfied with the authors' response. I agree with the authors that state-wise constrained RL problems are a class that deserves separate treatment and the authors' simple approach to tacking this challenging problems brings value to the community. Therefore I am revising my score to recommend for acceptance.

---

### Author Rebuttal · Authors · 2023-08-09

We sincerely thank all reviewers for all the detailed and helpful reviews!

We have revised the paper to address all the important issues raised in the reviews. Among all the reviews, there are some important questions about our paper we would like to highlight the answers here.


**Q1**: Alternative Proof for Theorem 1.

**A1**: We reworked the proof of Theorem 1, the basic idea is to first get the upper bound of discounted cumulative cost for finite horizon MDP (discussed in Appendix B), then we show the upper bound of non-discounted sum for finite horizon can be obtained via taking the limits of discounted sum version result. Thus, we unify the partial proofs of Theorem 1 and Theorem 2 which means this version of proof is more clear and more accurate than the existing one.

**Q2**: The novelty of our work.

**A2**: We acknowledge the extensive research that has been conducted on Constrained Markov Decision Processes (CMDPs), leading to the development of numerous safe Reinforcement Learning (RL) techniques, including well-known methods like Constrained Policy Optimization (CPO), Projected Constrained Policy Optimization (PCPO), Constrained Relative Policy Optimization (CRPO), and the application of Lagrangian methods, among others. These approaches have effectively dealt with the challenge of bounding cumulative costs within the CMDP framework. However, it is important to recognize that our work revolves around a distinct and novel problem landscape.

Our focus centers on addressing the intricacies of Safe Constrained Markov Decision Processes (SCMDPs), where safety constraints manifest instantaneously rather than cumulatively. This fundamental deviation significantly impacts the problem's complexity and the suitability of existing solutions. It renders all previous methodologies and theoretical insights, meticulously tailored for CMDPs, inapplicable to our SCMDP formulation.

Transitioning to our solution, the Maximally Constrained Markov Decision Process (MMDP) emerges as a pioneering contribution by offering a novel perspective on ensuring expected state-wise safety compliance within the realm of policy optimization. Unlike the conventional approaches such as CPO and its counterparts that hinge on the discounted summation over infinite horizon trajectories, MMDP leverages non-discounted summation over finite horizon Markov Decision Processes to maintain the up-to-now maximum cost. This distinction directly invalidates the proofs derived from CPO, making them unsuitable for the MMDP framework. Consequently, any theoretical findings within the MMDP context necessitate a distinct, dedicated derivation process. Our work thus leads the way in exploring trust region-based techniques within the realm of non-discounted MDPs and finite horizon scenarios.

Furthermore, our approach markedly diverges from state augmentation techniques employed in works like those by Sootla and Calvo-Fullana. While SCPO does utilize state augmentation to track the maximum incurred cost throughout the MMDP process, this element serves as a foundational tool rather than the primary innovation of SCPO. Substantial disparities exist between our approach and that of Sootla and Calvo-Fullana in terms of objectives and functionalities.

In summation, we confidently assert that our work tackles a substantially distinct problem domain compared to the existing body of research in safe RL. Our exploration of SCMDPs and the introduction of the MMDP framework usher in a new era of addressing instantaneous safety constraints within policy optimization.


**Acknowledgement**:

We are truly grateful for the invaluable suggestions you provided for our work. Your professional insights and recommendations hold immense significance as we embark on further revisions of the paper. Your careful review and guidance have been instrumental in shaping our research, and we genuinely appreciate your dedicated effort. Please know that your expertise is of great importance to us, and we welcome any additional questions or feedback you may have. Once again, thank you for your invaluable contribution.

---

### Decision · Program_Chairs · 2023-09-21

**Decision:**

Reject

**Comment:**

This paper introduces a novel framework called Maximum MDP to tackle the safe RL problem. It transforms the cumulative constraints in CMDP into a state-wise constrained optimization problem and provided some theoretical results on how to tune the constraint threshold to satisfy constraints during training.

On the overall, the reviewers think this paper is on the borderline. I also took a look at this work. While this paper presented a simple algorithm to tackle the CMDP problem, and the results are potentially promising, I find this paper currently unready for publication. My main concerns are: 1) the theoretical contribution only discusses constraint satisfaction but not sub-optimality performance and is similar to CPO; 2) In related work and experimental comparisons it seems the current version misses some relevant work (such as the Lyapunov approach); 3) the experimental results do not show strong improvements over other safe RL methods such as CPO or PCPO.

Unfortunately this field is more developed than it was a couple years ago, therefore more baseline comparisons and discernible performance improvement demonstrations are needed to justify the contribution of this work.